# Competing instabilities reveal how to rationally design and control active crosslinked gels

Bibi Najma[1,5], Minu Varghese[1,2,5], Lev Tsidilkovski[1], Linnea Lemma[1,3,4], Aparna Baskaran [1] & Guillaume Duclos [1]✉

How active stresses generated by molecular motors set the large-scale mechanics of the cell cytoskeleton remains poorly understood. Here, we combine experiments and theory to demonstrate how the emergent properties of a biomimetic active crosslinked gel depend on the properties of its microscopic constituents. We show that an extensile nematic elastomer exhibits two distinct activity-driven instabilities, spontaneously bending in-plane or buckling out-of-plane depending on its composition. Molecular motors play a dual antagonistic role, fluidizing or stiffening the gel depending on the ATP concentration. We demonstrate how active and elastic stresses are set by each component, providing estimates for the active gel theory parameters. Finally, activity and elasticity were manipulated in situ with light-activable motor proteins, controlling the direction of the instability optically. These results highlight how cytoskeletal stresses regulate the self-organization of living matter and set the foundations for the rational design and optogenetic control of active materials.

Active matter describes out-of-equilibrium materials composed of motile building blocks that convert free energy into mechanical work[1]. The continuous input of energy at the particle scale liberates these systems from the constraints of thermodynamic equilibrium, leading to hydrodynamic instabilities not found in passive materials[2–7]. Living cells are prototypical examples of adaptive multifunctional active materials[8]. They self-organize in diverse out-of-equilibrium states using the same molecular machinery, from spontaneously flowing vortices during cytoplasmic streaming[9] to polar asters during chromosome segregation[10]. Cells can spontaneously and reversibly trigger transitions between these non-equilibrium states by regulating the expression of their molecular building blocks. Understanding how these non-equilibrium dynamical transitions are driven at the molecular level is challenging. In particular, little is known about how to relate the forces generated at the microscopic scale to macroscopic emergent dynamics - both numerically and experimentally[11–17]. This is partly due to the non-linear collective organization of the force-generating units inherent to active matter. Consequently, most of the parameters conventionally used in symmetry-based hydrodynamic theories[1,18,19] are challenging to estimate, and they have not been related to the properties of the microscopic constituents. As a result, the predictive assembly of active matter with controllable material properties remains unexplored, despite recent advances in machine-learning-driven forecasting of active nematics[20,21], light-harvesting molecular motors[22–24], and the spatiotemporal control of active dynamics[25–27].

Transitions between active gels and active fluids have been quantitatively described in contractile actomyosin networks using a combination of experiments, theory, and simulations[17,28–31]. In particular, systematic investigation of the role of myosin-driven contractile stresses and passive actin crosslinkers has shown that the emergent fluid-like or solid-like properties of the network—in particular the

[1]Department of Physics, Brandeis University, Waltham, MA 02453, USA. [2]Department of Physics, University of Michigan, Ann Arbor, MI 48109, USA. [3]Department of Physics, University of California at Santa Barbara, Santa Barbara, CA 93106, USA. [4]Present address: Department of Chemical and Biological Engineering, Princeton University, Princeton, NJ 08544, USA. [5]These authors contributed equally: Bibi Najma, Minu Varghese. ✉e-mail: gduclos@brandeis.edu

ability to contract locally versus globally—are controlled by both the active stresses and the network connectivity[17]. However, it is unclear if extensile motor-polymer networks, in particular bundles of microtubules (MTs) and kinesin motors, behave in a similar fashion.

In this article, we show that investigating spontaneous deformations in a minimal in vitro system composed of cytoskeletal proteins can reveal how to rationally design and control active biomimetic materials. We combine experiments and theory to explore the origin of activity-driven instabilities in thin extensile networks composed of MT bundles, kinesin molecular motors, and crosslinkers. We show that depending on the molecular composition of the network, the extensile nature of the kinesin-MT bundles can drive two distinct instabilities: a bend instability, where the network spontaneously deforms in-plane, and a buckling instability, where the active network spontaneously buckles out-of-plane. We note here that both instabilities have been reported separately, (in-plane[32,33], out-of-plane[34,35]). These papers mostly focused on investigating the wavelength selection mechanism. Further, these experiments were all performed under different molecular compositions (different MT lengths, different molecular motors, different protein concentrations), which limits our understanding of why some active ordered materials bend in-plane or out-of-plane. The work presented here unifies these previously disconnected observations and demonstrates that both instabilities are generic features of active nematic elastomers. Of note, recent experiments and theory reached similar conclusions, confirming that MT gels undergo an ATP-dependent gelation transition[36]. However here, the important and distinct takeaway is that we leveraged the competition between these distinct unstable modes as a rheometer to connect the emergent macroscopic mechanical properties of the active network to the properties of its microscopic constituents.

A hydrodynamical model of a thin active elastomer shows that the balance between activity and elasticity controls the directionality of the most unstable mode: in the high activity low elasticity regime, active gels spontaneously bend in-plane[37] while in the low activity high elasticity regime, they buckle out-of-plane[34,35]. The transition where the instability changes direction is set by the ratio of active motors to passive crosslinkers. Using an enzyme kinetics model, we demonstrate how the activity and the elasticity of the network are set by the concentrations of molecular motors and crosslinkers. We show that motor proteins have a dual antagonistic role depending on ATP concentration: ATP-powered motor stepping fluidizes the crosslinked network, but between two power strokes, motors can also act as crosslinkers, enabling the transmission of elastic stresses[38]. The mapping between the cytoskeletal composition and the mechanical properties of the material provides a quantitative estimate of the active and elastic stresses for a broad range of molecular compositions. This multiscale framework enables the rational design of active biomimetic materials, which has been a long-lasting goal for the material genome initiative[39]. Finally, we show how the mechanical properties can be controlled in situ using light-dimerizable molecular motors and spatiotemporal illumination patterns. Taken together, these results serve as a paradigm for the rational design and control of active matter, a requirement for any adaptive and reconfigurable materials applications.

## Results

### Thin extensile active networks spontaneously deform in-plane or out-of-plane depending on their molecular composition

While ordered fluids at thermodynamic equilibrium tend to minimize free energy by uniformly aligning their elongated units, active nematics are intrinsically unstable[40]. The energy injected at the particle scale drives the spontaneous growth of long-range deformations[37] and subsequent nucleation of topological defects[41,42]. Here we assemble an active material composed of rod-like MTs and kinesin molecular motors (Fig. 1a). Stabilized MTs are either bundled by a depletant or crosslinked by PRC1, a MT-specific crosslinker[43,44]. The polymer

network is driven away from equilibrium by the adenosine 5'-triphosphate (ATP) fueled stepping of clusters of molecular motors. The motor clusters slide apart antiparallel MTs, exerting dipolar extensile stresses that drive the spontaneous growth of deformations, leading to chaotic dynamics[2,45] and active turbulence[46,47].

We confined this active material within a thin microfabricated channel (Fig. 1b). We initially flowed the material within the channel to uniformly align the polymer bundles along the channel length, similarly to passive rod-like colloids under shear flows[48,49]. Then, we stopped the shear flow, at which point the aligned active network was unstable, and periodic deformations with a finite wavelength spontaneously grew perpendicular to the initial alignment (Fig. 1c). Of note, the active material described above is a three-dimensional (3D) dilute suspension of flow-aligned crosslinked MT bundles. While the initial long-range nematic alignment is flow-induced, the complex fluid can be described as a 3D liquid crystal elastomer in the isotropic phase. This is distinct from the dense two-dimensional (2D) active nematic layer composed of MT bundles depleted onto an oil-water interface[41,46,50,51]. At high ATP concentrations, we observed the growth of in-plane deformations that result from the previously reported generic bend instability in extensile active liquid crystals (Fig. 1c and Supplementary Videos 1, 2, and 5)[32]. Decreasing the concentration of ATP led to an unexpected transition: at low ATP concentrations, in-plane deformations are suppressed (Supplementary Fig. 1), and regularly spaced patches of the network slowly grow out-of-focus (Fig. 1c and Supplementary Videos 3 and 4). Confocal microscopy confirmed that the out-of-focus domains correspond to MT bundles buckling out-of-plane (Fig. 1d, e and Supplementary Video 6). This instability is reminiscent of the out-of-plane buckling reported for suspensions of longer MTs and lower motor concentrations[34,35]. Eventually, the active network evolved into a chaotic flow regime, irrespective of the directionality of the first instability. It had been previously reported that the active bucking instability is accompanied by an anisotropic contraction of the network along the Z-axis when the MTs are bundled by a non-absorbing polymer[34,35]. While the periodic buckling is driven by the stepping of molecular motors, the reported contraction was a passive process driven by depletion forces. We did not observe any significant contraction along the Z-axis for the range of molecular composition corresponding to either the in-plane instability or the out-of-plane buckling (Supplementary Fig. 2). We did observe a slow contraction in the PRC-1 crosslinked network in the absence of activity (no molecular motors, or no ATP). In Fig. 1d, the active sheet is thinner than the channel height only where it buckles against the walls of the microfluidic channel (Supplementary Fig. 2d). In the regime explored here, the timescale associated with the contraction is larger than the characteristic time scale for the activity-driven instability. Therefore, the contraction does not impact the selection of the wavelength or the direction of the instability.

To further characterize the directionality of the instability, we defined the blurriness B as a scalar varying between 0 and 1 that quantifies the area fraction of the out-of-focus domains, as recently done in simulations of active nematic fluids[52]. Supplementary Fig 3 describes the quantitative procedure to segment the out-of-focus domains (see SI section 4.8 for details on the microscopy and image analysis algorithm, Supplementary Fig. 4, and Supplementary Video 7 for examples of segmented time series of various unstable networks). Briefly, if the instability is purely in-plane, then the whole image is in-focus and the blurriness is null ($B = 0$, diamond symbols on Fig. 1f). If the instability is purely out-of-plane, a large portion of the image is out-of-focus ($B > 0.5$, circle symbols on Fig. 1f). For intermediate values of blurriness, the instability is a superposition of in-plane and out-of-plane deformations (square symbols on Fig. 1f). Side by side quantitative comparison of confocal and epifluorescence imaging under various imaging conditions confirmed that the blurriness coefficient is

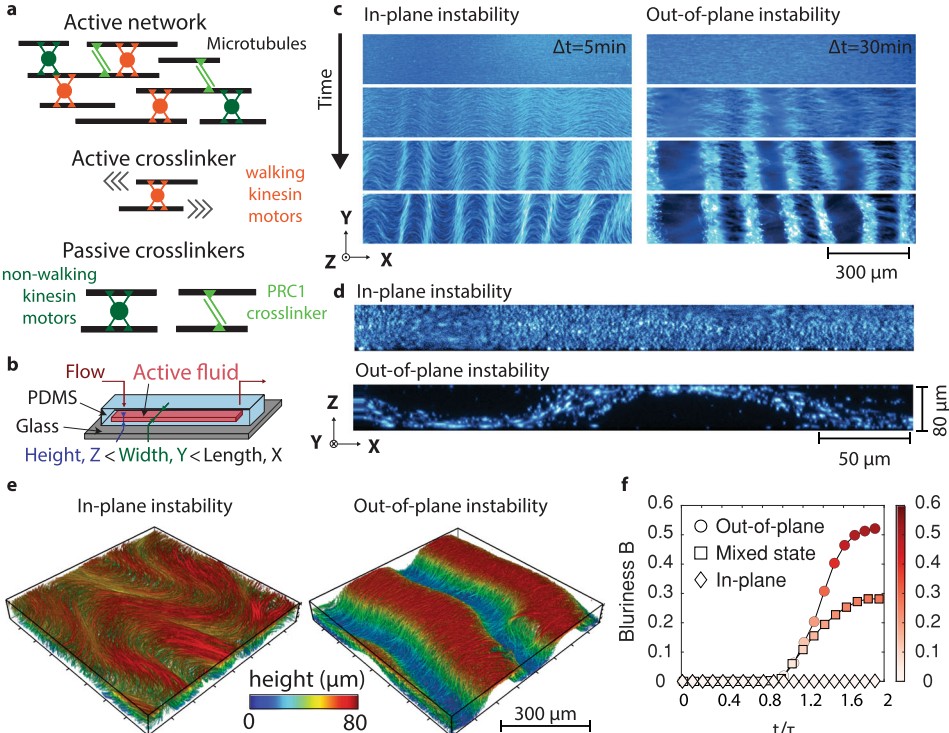

**Fig. 1 | Spontaneous growth of two instabilities in 3D active networks of crosslinked microtubules. a** Schematic of an active network: microtubules are crosslinked by clusters of kinesin-1 molecular motors and passive PRC1 cross-linkers. In the presence of ATP, they form stress–generating extensile bundles. In the absence of ATP, the molecular motors are not walking. Instead, they passively crosslink adjacent microtubules. **b** Schematic of the microfluidic channels with thin rectangular cross-sections ($H = 80\,\mu m$, $W = 3\,mm$, $L = 3\,cm$). **c** Time series for the in-plane and out-of-plane instabilities starting from flow-aligned networks ([motor cluster] = 20 nM and, respectively, [ATP] = 1.4 mM and 2 μM, fluorescent widefield microscopy). **d** Confocal cross-section along the direction of initial alignment of actively bending and buckling networks. Experiments were repeated $N = 5$ times. **e** 3D volume rendering of confocal images for the in-plane and out-of-plane instabilities. The color code shows optical sections taken at different depths. **f** Time evolution of the blurriness coefficient $B$ measured for three unstable networks. $\tau$ is the characteristic time for the growth of instability. The blurriness quantifies the fraction of the widefield fluorescent image that is out-of-focus. When the blurriness is null, the entire image is in focus and the instability is purely in-plane. The larger the blurriness, the more out-of-plane buckling is detected. The color map indicates the blurriness $B$. Source data are provided in the Source data file.

a robust metric of the escape of the extensile bundles into the third dimension (Supplementary Fig. 4)

We hypothesize that the ATP-controlled transition from out-of-plane buckling to in-plane bending is a signature of the softening of the active network. Multiple clues motivate this hypothesis. First, active hydrodynamic theory has shown that the in-plane bend instability is a generic feature of 3D active nematic liquid crystal confined in thin channels[4,32]. Second, motor clusters will also bind to MTs in the absence of ATP[53]. Decreasing the concentration of ATP leads to motors dwelling longer on MTs between two ATP-fueled power strokes, resulting in a larger proportion of motors passively crosslinking MTs instead of sliding them apart. Third, recent rheology experiments showed that the elastic and loss moduli of isotropic kinesin–MT networks depend on the concentration of ATP[38]. Importantly, these bulk rheology measurements were performed on active networks identical to the ones under study here (same MT length distributions and same kinesin-1 motor clusters); Finally, the out-of-plane buckling is reminiscent of the compressive winkling of thin elastic sheets[54].

## The balance between activity and crosslinking controls the directionality of the most unstable mode

To explore this hypothesis, we further investigated how the molecular composition of the active network controls the direction of the instability. Slowly increasing the ATP concentration triggered a sharp transition from an out-of-plane to an in-plane instability around 5 μM (Fig. 2a). Measurements on passive visco-elastic gels composed of biopolymers suggest that increasing the number of crosslinkers should lead to stiffer networks[55]. In agreement with this expectation, we observed that sparsely crosslinked networks spontaneously bent in plane while densely crosslinked networks buckled out-of-plane (Fig. 2b). When ATP was abundant, increasing the number of motor clusters induced a transition from an out-of-plane buckling to an in-plane instability (Fig. 2c). Interestingly, in the ATP limiting regime, increasing the number of motor clusters triggered a re-entrant transition from in to out-of-plane deformations (Fig. 2c). Finally, increasing the length of the MTs while maintaining a constant number of tubulin monomers led to a transition from in-plane instability to an out-of-plane bucking (Fig. 2d). Figure 2e summarizes these observations, suggesting that the ratio of active motors kinesin) to passive cross-linkers (PRC1 and non-ATP-bound kinesin) controls the directionality of the instability. Of note, we observed similar phase behaviors for active MT networks bundled by a depleting agent instead of a cross-linker (Supplementary Fig. 5) and for networks powered by non-processive K-365 kinesin motor clusters that still slide bundles apart but detach from the MTs after each step (Supplementary Fig. 6).

## Hydrodynamic model for an active elastomer

To better understand how the interplay between activity and elasticity sets the direction of the instability, we modeled the active network as a thin sheet of active nematic elastomer in a quasistatic medium[1,34,56]. We chose to model the network as a 2D material because the thickness of the network is always smaller than the other dimensions ($\tau = 80\,\mu m$, width = 3 mm, length > 2 cm) and because the thickness of the active sheet just before the instability does not vary for the range of

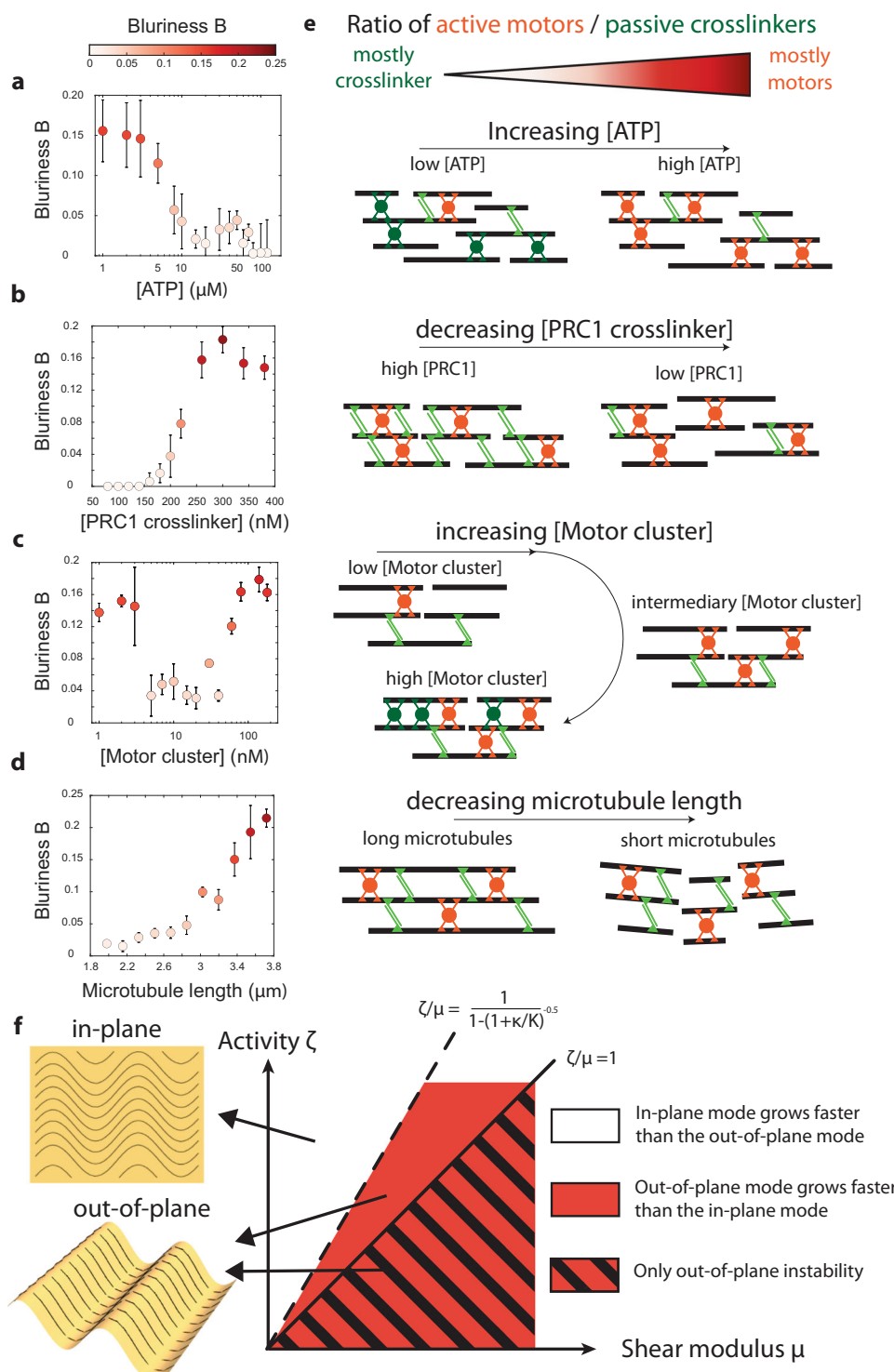

**Fig. 2 | The transition from an in-plane to out-of-plane instability is set by the ratio of active motors to passive crosslinkers.** Blurriness $B$ as a function of **a** ATP concentration: increasing [ATP] leads to a transition from an out-of-plane ($B > 0$) to an in-plane ($B = 0$) instability ([motor cluster] = 30 nM, [PRC1 crosslinker] = 100 nM); **b** PRC1 crosslinker concentration: increasing [PRC1 crosslinker] leads to a transition from an in-plane to an out-of-plane instability ([ATP] = 40 μM, [motor cluster]=60 nM). **c** Motor clusters concentration: increasing [motor clusters] can induce a re-entrant transition when [ATP] is low ([ATP] = 15 μM, [PRC1 cross-linker] = 100 nM). **d** Average microtubules' length ([ATP]=8 μM, [motor cluster] = 15 nM, [PRC1 crosslinker] = 100 nM). In **a**–**d**, the color map indicates the mean

blurriness $B$, and the error bars represent the standard deviation of the blurriness over at least three independent replicates. **e** Schematics of an active network where the ratio of active motors to passive crosslinkers or the polymers' length are changed. Orange motor clusters are ATP-bound (aka active), while green motor clusters are passively crosslinking microtubules. **f** Theoretical phase diagram showing how the interplay between activity $\zeta$ and shear modulus $\mu$ of a thin active elastomer sheet sets the direction of the most unstable mode. The sheet spontaneously deforms in 3D along the direction of the most unstable mode. In-plane deformations spontaneously grow only for $\zeta/\mu > 1$ (continuous line) while the out-of-plane modes are always unstable. Source data are provided in the Source data file.

molecular compositions studied here (Supplementary Fig. 2). The 2D hydrodynamic model can be derived from a 3D model by integrating over the thickness of the sheet (see SI section 3.3). We assumed that the sheet initially lays flat in the $xy$ plane with nematic order along $\hat{\mathbf{x}}$, and considered small deformations of the form $\vec{u} = (u_x, u_y, h)$. Further, we assumed that $\vec{u}$ varies spatially only along $\hat{\mathbf{x}}$. Since the material points of the elastic sheet are the nematogens, the fluctuations in the director are coupled to the displacement field in the sheet. As a result, the nematic director in the deformed state can be written as $\vec{n} = \hat{\mathbf{x}} + \partial_x \vec{u}$. The flat configuration of the sheet is unstable because a deformation $\vec{u}(\vec{r})$ at a point $\vec{r}$ on the sheet experiences destabilizing active forces $\zeta \vec{\nabla} \cdot (\vec{n}\,\vec{n}) = \zeta \left( \partial_x^2 u_y \, \hat{\mathbf{y}} + \partial_x^2 h \hat{\mathbf{z}} \right)$ from its surroundings[37]. The total free energy associated with the deformations is:

$$\mathcal{F}[u_x, u_y, h] = \frac{1}{2} \int dx dy \left[ \nu \left( \partial_x u_x \right)^2 + \mu \left( \partial_x u_y \right)^2 + \kappa \left( \partial_x^2 h \right)^2 + K \left[ \left( \partial_x^2 u_y \right)^2 + \left( \partial_x^2 h \right)^2 \right] \right]$$

(1)

where $\nu$ is the effective bulk modulus, $\mu$ is the shear modulus, $\kappa$ is the effective bending modulus, and $K$ is the nematic elasticity ($\nu$ and $\kappa$ are modified from their isotropic values due to the presence of nematic order, see SI section 1). Therefore, the deformation experiences a restoring force $-\frac{\delta \mathcal{F}}{\delta \vec{u}(\vec{r})}$ from the rest of the elastic sheet. It also experiences an additional restoring force $-\gamma \partial_t \vec{u}(\vec{r})$ through friction internal to the material and from the surrounding fluid. Assuming the system is overdamped, the balance of active, elastic and frictional forces experienced by a material point gives the following dynamics for the in-plane $u_y$ and out-of-plane $h$ deformations:

$$\partial_t u_y = \frac{1}{\gamma} \left[ (\mu - \zeta) \partial_x^2 - K \partial_x^4 \right] u_y$$

(2)

$$\partial_t h = -\frac{1}{\gamma} \left[ \zeta \partial_x^2 + (K + \kappa) \partial_x^4 \right] h$$

(3)

Note that both equations are unstable when the activity $\zeta$ is large. Equation (2) in the absence of elasticity is $\partial_t \vec{u} = -\frac{1}{\gamma} \left[ \zeta \nabla \cdot \vec{n}\,\vec{n} + \nabla \cdot \sigma^p \right]$ where $\sigma^p$ is the passive stress arising from the nematic deformations. Thus, it describes the active flow induced in the presence of strong substrate friction in the well-studied active nematic theories[1,56,57]. Equation (2), therefore, leads to an in-plane instability known as the generic instability in active nematics[37]. Equation (3) on the other hand describes height fluctuations in an elastic thin film and leads to an activity-driven out-of-plane buckling instability, reminiscent of Euler buckling in solids[34,35]. Equation (3) shows that out-of-plane modes are always unstable at any non-zero activity, while Eq. (2) predicts that in-plane modes are unstable only when the activity is larger than a critical activity $\zeta^* = \mu$, where $\mu$ is the shear modulus. The existence of a non-zero critical activity is a consequence of the elastic response of the active sheet, which is fundamentally different from critical activity resulting from either confinement[32] or friction[51]. Of note, surface tension can be neglected despite the fact that confinement could confer a frame tension. Indeed, the surface tension corresponds to the lowest value of activity at which the out-of-plane instability is observed, which is about 40 times smaller than the lowest stretching modulus which corresponds to the lowest activity at which the in-plane instability is observed.

This theory predicts a phase diagram composed of three distinct regimes (Fig. 2f):

i)  the active sheet buckles out-of-plane because only out-of-plane modes are unstable: $0 < \zeta/\mu < 1$;
ii) the active sheet buckles out-of-plane because the most unstable out-of-plane mode grows faster than the most-unstable in-plane mode: $1 < \zeta/\mu < \frac{1}{1-(1+\frac{\kappa}{K})^{-0.5}}$ where $\kappa$ is the bending modulus and K is the nematic elasticity;

iii) the active sheet bends and stretches in-plane because the most unstable in-plane mode grows faster than the most-unstable out-of-plane mode: $\zeta/\mu > \frac{1}{1-(1+\frac{\kappa}{K})^{-0.5}}$

This minimal model demonstrates that the direction of the instability is set by the balance between active stresses and passive elastic stresses. Of note, both in and out-of-plane instabilities are possible in the original active nematic theory[37,58] and have been reported in hydrodynamic simulations of purely extensile active nematic fluids[52]. However, the out-of-plane component of the generic bend instability grows slower that the in-plane component when the active nematic fluid is confined in a thin channel[32]. We are therefore confident that the bucking instability is a signature of the elasticity of the network. Finally, our theory does not display a fluid-to-solid transition since the shear modulus $\mu$ increases continuously when the instability changes direction. It only becomes the theory of a fluid when $\mu$ vanishes. However, it confirms that increasing the elasticity of a thin active gel leads to a transition from the generic bend instability to a buckling instability. It is consistent with both our experimental observations and recent bulk rheology experiments on the same system[38].

## A reaction kinetics model connects molecular composition to the activity and mechanical properties of the network

We developed a model based on Michaelis–Menten enzyme kinetics to connect the concentration of cytoskeletal proteins to the macroscopic material parameters that determine the dynamics of the active network. Motor proteins can either generate forces or act as crosslinkers depending on ATP concentration. Here, this simple model estimates the number of active motors (ATP-bound motors stepping on MTs) and the number of passive motors (non-ATP-bound motors that passively crosslink the network, see SI for details). Extensile stresses are generated by the relative sliding of MTs past one another[2]. Therefore, we assumed that the activity $\zeta$ is proportional to the elongation rate of a MT bundle, which itself depends linearly on the number of ATP-bound motor clusters sliding the MTs by the Michaelis–Menten relation (see SI section 2.1):

$$\zeta = \zeta_0 \cdot \frac{[\text{ATP}] \cdot [\text{Motor clusters}]}{\frac{k_h + k_-}{k_b} + [\text{ATP}]}$$

(4)

where $\zeta_0$ is an activity constant that depends on the efficiency with which ATP hydrolysis translates into mechanical work, $k_h$ is the ATP hydrolysis rate, $k_-$ and $k_b$ are, respectively, the unbinding and binding rates of ATP to kinesin motors[59]. Next, bulk rheology experiments on passive isotropic networks of crosslinked biopolymers above the isostatic transition showed that the shear modulus $\mu$ and the bending modulus $\kappa$ vary as $\mu = \mu_0 [\text{crosslinkers}]^2$ and $\kappa = \kappa_0 [\text{crosslinkers}]^2$ (see SI section 2 for a discussion of the exponent and the potential impact of orientational order)[55,60]. Finally, we estimated the total number of crosslinkers as follows: $[\text{crosslinkers}] = p_0 [\text{PRC1}] + [\text{motor clusters}]_{\text{total}} - [\text{motor clusters}]_{\text{stepping}}$

PRC1 being a sensitive protein to purify, we considered that only a fraction $p_0$ of the proteins is active. The last two terms on the right-hand side correspond to the fraction of molecular motors that cross-link MTs instead of sliding them apart.

Combining the hydrodynamic model with the enzyme kinetics, we derived a theory with four unknown parameters: (i) the ratio of $\zeta_0$ and $\mu_0$, (ii) the ratio of $K_0$ and $\kappa_0$, (iii) the ratio of $K_0$ and $\zeta_0$, and (iv) $p_0$ (see SI section 2.4 for the equation of the phase boundary). We performed over $N = 1000$ experiments to build four experimental phase diagrams showing how the transition from in-plane bending to out-of-plane buckling depends on ATP, motor clusters, and cross-linker concentrations, and the average length of the MTs (Fig. 3). The theoretical model recovers that the more crosslinked the network is,

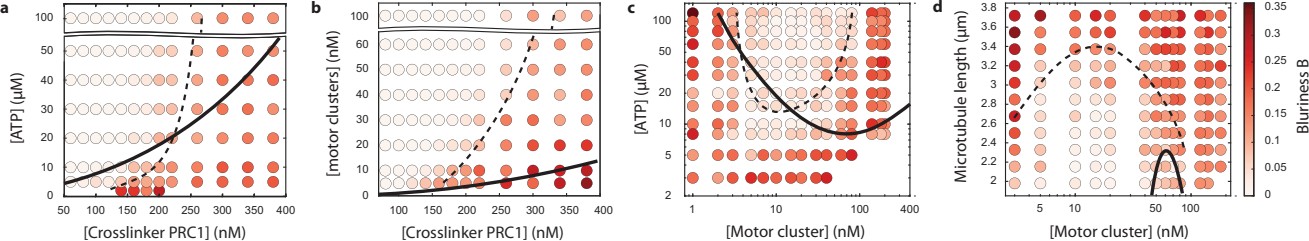

**Fig. 3 | Biochemical processes control the transition from a bend instability to a buckling instability.** Experimental phase diagrams, experimental phase boundaries (dashed lines) and theoretical phase boundaries (continuous black lines) describing the transition from an in-plane fluid-like bend instability to an out-of-plane solid-like buckling for various **a** ATP and PRC1 crosslinker concentrations ([motor clusters] = 60 nM), **b** motor clusters and PRC1 concentrations ([ATP] = 1

mM), **c** ATP and motor clusters concentrations ([PRC1] = 100 nM), and **d** microtubules' length and motor cluster concentration ([ATP] = 8 μM, [PRC1] = 100 nM). The color map indicates blurriness $B$ and is the same for the four phase diagrams. Each colored circle corresponds to a distinct experimental realization. Source data are provided in the Source data file.

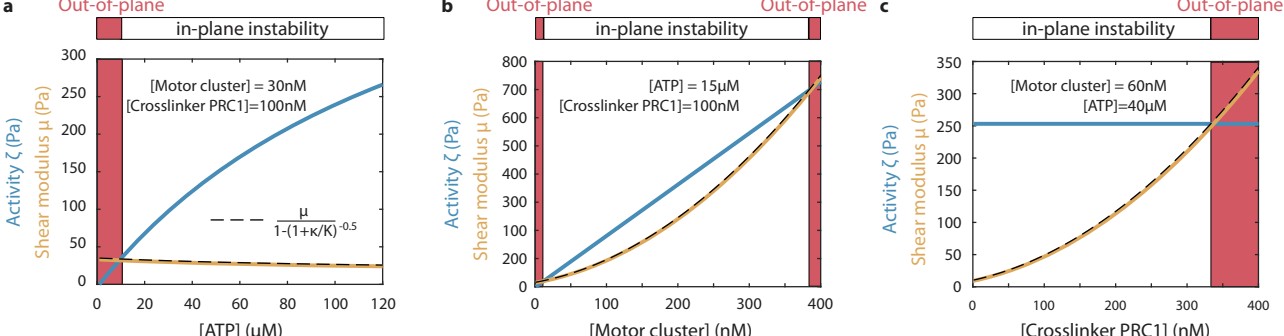

**Fig. 4 | Molecular composition of the network sets its activity and elasticity.** Activity (blue curves) and shear modulus (orange curves) when varying **a** ATP concentration, **b** motor cluster concentrations, and **c** PRC1 crosslinker concentrations. Active and elastic stresses are inferred from fitting the theoretical model to the experimental phase diagrams (as described in the SI sections 2.4 and 3). When

the activity is above (resp. below) the dashed line, the network bends in-plane (resp. buckles out-of-plane). $\mu_0$ is estimated from bulk rheology experiments by the authors from ref. 38 (see SI section 3.4), while the ratios $\zeta_0/\mu_0$, $K_0/\kappa_0$, and, $\kappa_0/\zeta_0$ are fitted parameters.

the more activity is required to soften the network (Fig. 3a, b). The model also captures the re-entrant transition controlled by motor clusters concentration: increasing the number of motors first softens but then induces a rigidification of the network as adding more motors in the ATP limiting case is equivalent to increasing the number of crosslinkers (Fig. 3c). Finally, the model includes the influence of the length of the MTs through its impact on the nematic elasticity, which qualitatively describes the transition from a bend instability for short MTs to a buckling instability for longer MTs (Fig. 3d). The out-of-plane wavelengths are also well captured by the theory (Supplementary Fig. 7).

Careful fitting of the four phase diagrams and the corresponding wavelengths to the active gel theory using a Markov Chain Monte Carlo method allows estimating the fraction of active PRC-1 ($p_0 \sim 60\%$) and the three unknown ratios $\zeta_0/\mu_0$, $K_0/\kappa_0$, and $\kappa_0/\zeta_0$ from the 3D active gel theory (see SI section 3 for fitting procedure). $\mu_0$ was estimated from bulk rheology of 3D ATP-depleted networks of MTs and kinesin motors by the authors from ref. 38 (SI section 3.4). As a result, we can infer the magnitude of the 3D active stresses for a broad range of molecular compositions, which has been an elusive milestone in the field of biomimetic active matter. Here, the quasi-quantitative agreement between theory and experimental data provides a multiscale map between the activity and shear modulus of the network and its molecular composition (Fig. 4). We estimated the activity to be around 10–1000 Pa for the range of motor proteins and ATP concentrations usually reported in the literature for this MT-based 3D network[2,38,44,61,62] (see SI section 3.5 for comparison with estimates of the activity from the literature). This mapping reveals how molecular-scale chemical reactions between the motor proteins and the MTs govern the

rheology and emergent dynamics of a biomimetic material. Further, it provides guidance for the rational design of cytoskeletal-based active matter in targeted out-of-equilibrium states.

While this model describes well the re-entrant phases, the range of motor cluster concentrations where the fluid phase is observed is slightly different in the experiments and in the theory ([motor cluster]$_{exp}$ = 20 nM while [motor cluster]$_{th}$ = 70 nM for the extrema of the fluid–solid phase boundary in Fig. 3c, d). This discrepancy might result from an ambiguity of the proper scaling for the elasticity of a nematic elastomer composed of MTs (SI section 2.2). Of note, the ratio of motors to crosslinkers required to fluidize the network is around 10%. In the regime explored here, each MT is decorated on average by 0.2–25 motor clusters and 10–80 crosslinkers[63]. Interestingly, increasing ATP induces a softening of the network (Fig. 4a). A similar effect has been observed for the nematic elasticity of a MT-based 2D active nematic liquid crystal[20,50].

### Optogenetic control of the molecular motors sets the direction of the most unstable mode

We further demonstrate how to leverage light-activable molecular motors to optogenetically control the activity and elasticity of the network in situ. We replaced the conventional motor clusters with light-dimerizable kinesin-1 motors[24,64]. In the absence of light, the motors do not dimerize: they hydrolyze ATP, step onto MTs, but do not induce any relative sliding (Fig. 5a). When exposed to blue light, the motors form a cluster that can slide apart adjacent MTs, much like the extension produced by the conventional kinesin-1 clusters. These light-dimerizable motors enable spatiotemporal control of motor activity. In particular, we show that tuning the intensity and

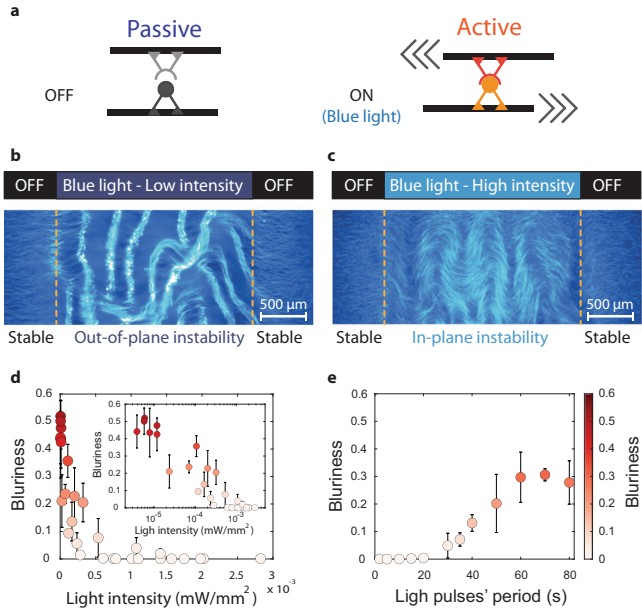

**Fig. 5 | Optogenetic control of the direction of the instability. a** Schematics of the light-dimerizable kinesin-1 motors. In the absence of light, the motors hydrolyze ATP, step on individual microtubules, but do not dimerize, which does not induce any relative sliding of the microtubules. When blue light is turned on (488 nm wavelength), the motors dimerize. Their stepping produces a net extension of the microtubule bundle. We shined pulsed patterns of blue light on a flow-aligned network; **b** for low blue light intensity, the network buckles out-of-plane; **c** for high blue light intensity, the network bends in-plane. In the absence of light, the network is stable and stays flow aligned; **d** Blurriness $B$ of the unstable active network for varying light intensity (main: lin–lin plot, insert: lin–log plot showing the sharp transition around 0.1 μW/mm³). **e** Increasing the blue light pulses' period induces a transition from in-plane bending to out-of-plane buckling for a critical pulse period $\tau \sim 30$ s. The color map is the same for **d, e** and indicates the blurriness $B$. Error bars in **d, e** represent the standard deviation of the mean over $N = 3$ independent replicates. Experiments shown in **b, c** were, respectively, repeated $N = 30$ and $N = 45$ times. Source data are provided in the Source data file.

the period of the activity pulses modulate the direction of the instability. First, the network was stable in the absence of light. Then, at low intensity, the network buckled out-of-plane (Fig. 5b). Increasing the light intensity induced a transition to an in-plane instability (Fig. 5c, d), which is consistent with the number of dimerized motors increasing and then saturating with the light intensity. The wavelengths of the instability were also consistent with an increase in activity (Supplementary Fig. 8). The direction of the instability can also be controlled by changing the period of the light pulses (Fig. 5e). Shining high-intensity pulsed light at high frequency triggered an in-plane instability while shining light at a low frequency triggered an out-of-plane buckling. The critical pulse period required to fluidize the network is around 30 s. While this transition cannot be captured by the quasistatic approximation of our model, it is consistent with the off-rate of the light-induced dimer $\tau \sim 30$ s[65]: if the exposure period is larger than 30 s, then only a fraction of the stepping motors is dimerized, which result in a lower activity. Finally, we note that deformations only partially relax once the activity is turned off, probably due to the presence of crosslinkers (Supplementary Fig. 9 and Supplementary Video 8). In the future, other control strategies, maybe with light-controllable crosslinkers, could be implemented to reversibly deform MT-based active materials in 3D.

## Discussion

To fully harness the engineering potential of active matter and trigger self-organization on demand, one needs to be able to both (a) design

materials with targeted activity and mechanical properties, and (b) control these properties in space and time. Here we have achieved both. First, we predictively designed coarse-grained active and elastic stresses from microscopic building blocks by combining quantitative measurements with theory at two levels—hydrodynamics and enzyme kinetics. As a result, we are able to infer the magnitude of the active stresses generated by kinesin motors for 3D active materials. While simplifications both in the scaling of the shear modulus and in the Michaelis–Menten kinetics neglect some of the molecular complexity of the cytoskeletal proteins, like the load dependence of kinesin stepping[59,66], this work leads the way to rationally design active materials with targeted mechanical properties.

It is interesting to compare these results to the large body of work on actomyosin gels. There is growing evidence that actin-myosin and kinesin–MT-based active matter can self-organize in similar out-of-equilibrium phases, from 2D extensile nematic liquid crystals[2,61] to 3D contractile networks[17,67]. However, despite these equivalent final states, the microscopic mechanisms that drive these hierarchically self-organized states appear fundamentally different. For example, myosin-mediated active stresses have been shown to dramatically stiffen crosslinked actin gels[28,29]. This is fundamentally different from what we report here for networks of kinesin and MTs where extensile active stresses effectively soften the crosslinked network. This suggests some fundamental differences in the microscopic origin of the emergent mechanics of contractile actomyosin gels and extensile kinesin-MT networks.

Previously, control of active matter focused mostly on 2D dynamics and relied on tuning the effective motility of topological defects in 2D active nematics[22] and in 2D polar active matter[24], or triggering local contraction of isotropic actomyosin gels[68]. Here, we identified and implemented spatiotemporal design principles to drive targeted deformations in 3D, carefully controlling the magnitude of the active and elastic stresses in situ. From a material science perspective, our work yields insights into developing reconfigurable active materials where transitions between non-equilibrium states can be triggered spontaneously or externally. From a life science perspective, this framework illustrates how cytoskeletal mechanics is controlled via spatiotemporal activity patterns. More generally, this work might shed light on how the spatiotemporal regulation of a finite pool of proteins can soften or rigidify the cell cytoplasm to drive collective flows while transmitting forces within a cohesive tissue in vivo. For example, during gastrulation, cells in the ventral tissue coordinate pulses of actomyosin contractility to drive a solid-to-fluid transition, triggering the out-of-plane deformation of a 2D tissue[69–71]. Contrarily, during vertebrate body axis elongation[72], tissues undergo a timed fluid-to-solid jamming transition that promotes tail bud formation. Our work suggests that the spatiotemporal control of the activity and the mechanical properties of the cytoskeleton is a generic physical mechanism that governs how living and active matter change shape in 3D.

## Methods

### Protein purification protocols

**Microtubules.** Tubulin dimers were purified from bovine brains through two cycles of polymerization–depolymerization in high molarity PIPES (1,4-piperazindiethanesulfonic acid) buffer[73]. Fluorophore-labeled tubulin was prepared by labeling the purified tubulin with Alexa-Fluor 647-NHS (Invitrogen, A-20006)[42]. GMPCPP (Guanosine 5′-(α,β, methylenetriphosphate)), a non-hydrolyzable analog of GTP was used to stabilize the dynamic instability in the MTs. Polymerization mixture consisted of 80 μM tubulin (with 3% fluorescently labeled tubulin), 0.6 mM GMPCPP and 1 mM dithiothreitol (DTT) in M2B buffer (80 mM PIPES, 1 mM EGTA, 2 mM MgCl₂). After adding all the components, the mixture was incubated at 37 °C for 30 min, and subsequently for 6 h at room temperature (20 °C). This

method resulted in MTs of 1.5 μm length[41]. The stock concentration was 8 mg/mL. MTs were aliquoted in small volumes (10 μL), flash-frozen in liquid nitrogen, and stored at −80 °C.

**Crosslinkers—PRC1-NSΔC.** The truncated PRC1-NSΔC (MW: 58 kDa), a MT crosslinking protein, was expressed and purified in Rosetta BL21(DE3) cells using an established protocol described elsewhere[43]. The proteins were flash frozen with 40% sucrose and stored at −80 °C. The final concentration of PRC1 was measured by a Bradford assay.

**Molecular motors—K401 and K365 motors.** K401-BIO-6xHIS (processive motor, dimeric MW: 110 kDa) and K365-BIO-6 xHIS (non-processive motor, MW: 50 kDa) are biotinylated kinesin constructs derived from N-terminal domain of *Drosophila melanogaster* kinesin-1, truncated at residue 401 and 365, respectively, and labeled with six histidine tags. The motor proteins were transformed and expressed in Rosetta (DE3) pLysS cells and purified following established protocols described previously[74]. The proteins were stored in a 40% wt/vol sucrose solution at −80 °C. The final concentration of kinesin was measured by Bradford assay.

We used tetrameric streptavidin (ThermoFisher, 21122, MW: 52.8 kDa) to assemble clusters of biotinlabeled kinesins (KSA). To make K401-streptavidin clusters, 5.7 μL of 6.6 μM streptavidin was mixed with 5 μL of 6.4 μM K401 and 0.5 μL of 5 mM DTT in M2B. This mixture was incubated on ice for 30 minutes. K365-streptavidin clusters were prepared by mixing 5.7 μL of 6.6 μM streptavidin, 3.1 μL of 20 μM K365, 0.5 μL of 5 mM DTT, and 1.94 μL of M2B and then left to incubate on ice for 30 min.

**Light-activable motors: K365-iLID and K365-micro.** K365-iLID and K365-micro were designed by Linnea Lemma and Tyler Ross and were purified following the protocol described in ref. 24. Briefly, two chimeras of the *D. melanogaster* kinesin K365, namely, K365-iLID and K365-micro, were expressed in *Escherichia coli* Rosetta 2(DE3)pLysS cells and purified using ÄKTA pure FPLC system. MBP domain was cleaved with TEV protease (Sigma Aldrich). The proteins were snap-frozen in 40% glycerol and stored at −80 °C.

**Assembling 3D active network**
The networks are composed of:
- Alexa 647-labeled GMPCPP stabilized MTs with an exponential distribution of lengths with an average of 1.5 μm, unless otherwise specified,
- Multi-motor kinesin complexes self-assembled from tetrameric streptavidin and two-headed biotinylated kinesin (K401-Bio) or single-headed biotinylated kinesin (K365-Bio)[32],
- a specific MT bundling protein, PRC1 (the protein regulator of cytokinesis 1), that passively crosslink antiparallel MTs, but still allow interfilament sliding[43].

In the presence of ATP, these proteins form a network of extensile bundles. An ATP regeneration system (phosphoenol pyruvate (26 mM PEP, Beantown Chemical, 129745) and pyruvate kinase/lactate dehydrogenase enzymes (2.8% v/v PK/LDH, Sigma, P-0294) was used to sustain a constant ATP concentration. An oxygen scavenging system comprised of glucose (18.7 mM), DTT (5.5 mM), glucose oxidase (1.4 μM), and catalase (0.17 μM) was used to decrease photobleaching. Active network composed of MTs (1.3 mg/mL), ATP (1420 μM), and K401 clusters (120 nM) remain active for 6–8 h.

For some experiments (Supplementary Fig. 3), PRC1 was replaced by either 20 kDa PEG (polyethylene glycol) (0.8% (wt/vol) [Sigma] or Pluronic F-127 2% (wt/vol) [F-127, Sigma P2443. MW: 12.5 kDa].

For consistency, a large volume of premix was prepared, aliquoted, and snap frozen in liquid nitrogen for each phase diagrams shown in Fig. 3. Frozen MTs (stored at −80 °C) were thawed immediately before

use in an experiment. All the experiments were performed at room temperature.

**Flow chamber assembly**
All flow chambers have the same dimensions ($H = 80$ μm, $W = 3$ mm) and were assembled using two different approaches. No differences were observed between the two methods. The first method relies on a microfabricated channel made out of PDMS. The channel was prepared from a molding master created by machining cyclic olefin copolymer (COC)[75]. PDMS was poured over the master, and cured for 1 h at 70 °C. The channel was removed from the master, inlet and outlet were punched, followed by oxygen plasma bonding to a glass slide ($26 \times 75 \times 1$ mm). The channel was incubated with 2% Pluronic solution for 30 min to block any non-specific protein adsorption. The second method consists in two glass slides spaced by a layer of parafilm. The glass surfaces were coated with an acrylamide brush to resist non-specific protein adsorption onto the glass due to depletion[2]. Parafilm spacers were cut and placed between the two glass surfaces followed by a mild heat treatment at 60 °C to melt parafilm so it can bind to the glass surfaces. The active mixture was loaded into each channel type by capillarity and sealed with an UV-curing optical adhesive (NOA 81, Norland Products Inc.).

**Microscopy**
**Widefield microscopy.** The Alexa 647-labeled MT networks were imaged using an inverted wide-field microscope (Nikon Ti-E or Ti2) with a fluorescent filter (Semrock Cy5-4040C), a SOLA light engine (Lumencor), a ×10 objective (Nikon Pan Fluor, NA 0.3) and a CCD or a sCMOS camera (Andor Clara E, Hamamatsu orca flash 4.0). The illumination and the data acquisition were controlled by micro-manager (μManager, Version 2.0.0-gamma[76]). All the measurements were performed at room temperature.

**Confocal microscopy.** Confocal fluorescence images of MTs were obtained under Leica Application Suite X (LAS X) control on a laser scanning confocal microscope (TCS-SP8, Leica Microsystems GmbH) equipped with photomultiplier tubes. Fluorescence was excited with 638 nm laser diode for Alexa Fluor 647. For morphological analysis, a z-stack with a $775 \times 775$ μm field of view was scanned with a step size of 2 μm using a non-immersion ×20 objective (HCX PL Fluotar, numerical aperture, NA = 0.50).

**Reporting summary**
Further information on research design is available in the Nature Research Reporting Summary linked to this article.

## Data availability
All study data are included in the article and/or Supporting Information. Source data are provided with this paper.

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

## Acknowledgements
B.N. and G.D. acknowledge support from a NSF CAREER award DMR-2047119. We also acknowledge the use of the optical, microfluidics, and biomaterial facilities supported by NSF MRSEC DMR-2011846. M.V. acknowledge support of HFSP grant RGP0031/2020. M.V., L.S., L.L., and A.B. acknowledge support of NSF MRSEC DMR-2011846. Computational resources were provided by the Brandeis HPCC, which is partially supported by the NSF through DMR-MRSEC 2011846 and OAC-1920147. We thank the director of the Brandeis Biomaterial Facility, Shibani Dalal, for help with protein purification. We thank Tyler Ross for help with the opto-K365 constructs. We thank Guillaume Sarfati, Jean-Christophe Galas, Andre Estevez-Torres, Hannah Yevick, Peter Foster, Zvonimir Dogic, Michael Hagan, and John Edison for discussion.

## Author contributions
G.D. designed the research; B.N. performed the experiments; B.N. and G.D. analyzed the data; M.V. and A.B. developed the theoretical model; M.V., B.N., and L.T. fitted the data; L.L. contributed to the experiments with optogenetic molecular motors; G.D. and A.B. supervised the research; B.N., M.V., A.B., and G.D. wrote the manuscript.

## Competing interests
The authors declare no competing interests.

## Additional information

**Correspondence and requests** for materials should be addressed to Guillaume Duclos.

