## [Peer Review File · Nature Communications]

REVIEWER COMMENTS

Reviewer #1 (Remarks to the Author):

In this revised version, Dr. Duclos and colleagues have significantly improved the article. I completely agree with their assertion that understanding how the emergent mechanical properties of active gels depend on the properties of their stress-generating microscopic constituents is an important open challenge in active matter. That the authors attempt to answer this is indeed important and commendable. However, I still have questions about some parts of the article. I think these need to be fully clarified before I can judge the suitability of the article for this journal.

1. The authors say in this revised manuscript that "Of note, both in and out-of-plane instabilities are possible in the original active nematic theory (36, 57) and have been reported in hydrodynamic simulations of purely extensile active nematic fluids (51). However, the out-of-plane component of the generic bend instability grows slower than the in-plane component when the active nematic fluid is confined in a thin channel (32). We are therefore confident that the buckling instability is a signature of the elasticity of the network. "

I am not sure how 32 establishes that a buckling instability requires network elasticity. That is, I don't see how 32 establishes that a fluid 2d layer (the theory that the authors construct assumes a 2d layer; that is, it averages over the thickness of the gel) necessarily bends in-plane before buckling. Note that the character of the in-plane and out-of-plane instabilities are somewhat different (and, consequently, the boundary conditions on the director field are probably different). The in-plane instability is analogous to the one discussed in 32: the instability of an active fluid filling a confined channel beyond a critical activity. The out-of-plane instability is distinct from this. It is an instability (at least in their calculation) in which the gel layer buckles as a whole without any extra distortion within the layer. There is no reason for that to grow slower than the in-plane bend instability. To see this, consider the following toy model: a 2d fluid nematic membrane. The nematic order is along x . The y confinement is smaller than the active lengthscale (this direction is perpendicular to the membrane normal, which is along z); therefore, there is no in-plane bend instability at all. However, as essentially the authors' calculation here shows, the buckling instability is still inevitable in this case (inevitable, if we assume a tension-free membrane). The height fluctuations of the membrane are still slaved to director fluctuations, leading to a negative effective surface tension along x . The only stabilisation of height fluctuations is due to the

bending modulus, which appears at higher order in gradients (along x). Therefore, in this geometry, one has a buckling instability but no bend instability in a fluid system. The confinement along the z direction doesn't change this conclusion. Therefore, the blanket statement that the authors make is incorrect, and the buckling instability is not inherently a signature of the elasticity of the network.

2. In the response letter, the authors say "We removed any claim of the transition from bending to buckling correspondsto the fluid to solid transition in the revised manuscript (see updated section I.304-I.309)." Yet, in the main text, the authors say "We hypothesize that the ATP-controlled transition from out-of-plane buckling to in-plane bending is a signature of a solid-to-fluid transition. (I188-9)". Later, they write "We performed over $N=1000$ experiments to build four experimental phase diagrams showing how the transition from an active fluid to an active solid depends on ATP, motor clusters, and crosslinker concentrations, and the average length of the microtubules (Fig. 3). "

This seems contradictory. Possibly, the authors mean that they have removed all mention of solid-fluid transition in the theoretical section (which they have; they also correctly point out that there is no solid-liquid transition). However, the claims in the theoretical and experimental sections are somewhat contradictory. The claim in the experimental section is that there is a true solid-fluid transition, while the theory suggests that no such transition is actually necessary (the authors claim that this is consistent with all their observations). All one needs is the activity to be large enough or the shear modulus to be small enough. If that is the claim, the statements about the solid-fluid transition hypothesis have to be culled from other parts of the paper. Instead, one has the more consistent (but perhaps less spectacular) hypothesis that the shear modulus of an active elastomer changes with ATP. Note that these are, in principle, two distinct hypotheses and what sort of parameters controls the behaviour of the macroscopic gel depend on the hypothesis.

3. I don't know why the authors label the bend instability a fluid-like instability and the buckling a solid-like instability. The authors themselves show that the bend instability exists in a system with a finite shear modulus (by definition, a solid). It is also evident that the buckling instability could have affected a fluid layer (as discussed above). I find this nomenclature confusing.

4. I agree that the elastomer will be semi-soft and also that it is possible that the experimental regime is such that the semi-softness is not probed. I further agree that the surface tension (incidentally, it must be present and generically be non-zero -- since it is allowed by symmetry, its value cannot be 0 except at special points in the parameter space. Since the system is not tuned to a critical point with 0 surface tension, there will be a surface tension) may be small. However, the argument presented by the authors for the small value of the surface tension should be in the main text, and they should clearly say why they ignore surface tension in the main text.

5. The authors mention a theory involving the apolar order parameter Q_{ij} and the strain tensor u_{ij} in a footnote in the supplement. However, strangely, the energy they use for the nematic

elastomer has a term $\gamma(\nabla \cdot \mathbf{u})Q_{ij}u_{ij}$ but not a simpler term $\propto Q_{ij}u_{ij}$ in the free energy. This is not inconsequential. A look at PRE 66, 011702 shows that the director is indeed slaved to the strain field but $\Delta n_{\perp} \neq u_{\parallel}$ as the authors have but $\Delta n_{\perp} \propto u_{\parallel}$ where the proportionality constant depends on the value and the sign of the coefficient of the $Q_{ij}u_{ij}$ free-energy coupling. That is, $\Delta n_y \neq \partial_x u_y$ but $\Delta n_y \propto \tilde{\gamma} \partial_x u_y$ where $\tilde{\gamma}$ is the coefficient of the $Q_{ij}u_{ij}$ free energy term. While the final free-energy (only depending on the displacement field) that the authors write can still be obtained from the nematic elastomeric theory, the consequence of the non-trivial slaving of the director field to the displacement fluctuations are important for the active force density. In particular, the coefficient (and sign) of the planar active force density should depend on the combination $\zeta \tilde{\gamma}$. The authors essentially seem to tacitly set $\tilde{\gamma}=1$. This needs to be justified, especially since $\tilde{\gamma}$ can, in principle, have an arbitrary sign (this is related to my earlier question "seems to claim that extensile active nematic elastomers are always unstable to bend at sufficiently high activity and, conversely, contractile active nematic elastomers are always unstable to splay at sufficiently high activity. Is that really the case?").

6. The authors in the revised manuscript talk about elastic moduli "below the isostatic transition". I am confused by this. I thought that the elastic moduli vanish if the crosslink density is below the isostatic transition (there are zero modes and hence, the solid cannot resist any imposed stress).

7. While the authors make a serious and commendable attempt to obtain the values of the hydrodynamic parameters used in active gel theories from the properties of microscopic constituents, because of various uncertainties (even for the active coefficient), they essentially only succeed in obtaining order of magnitude estimates. Other articles have attempted to obtain such order-of-magnitude estimates before (with varied success). Therefore, to me, stressing this too much sounds a bit like overselling.

Reviewer #2 (Remarks to the Author):

In this resubmission, Nahma et al. have significantly improved the manuscript, and have addressed my comments and criticisms in a satisfactory manner. In particular, they have clarified the difference with earlier work on, apparently, the same system. They explain that earlier work had a disparity of material conditions, which precluded a unified view. Instead, they demonstrate that different phenomena reported earlier can be replicated in the same system by changing the ATP concentration. This allows to propose a theoretical model to loosely explain the phenomena and, more importantly, to estimate important material parameters. In particular, the idea that this work presents a sort of "rheometer" to probe active material properties is quite attractive.

As a minor detail, I think there is an error in panel 5e (which escaped me in the original manuscript). The X-axis has units of time (it is a time), not a “rate”.

Finally, a major issue.

The authors were honest in their initial submission to acknowledge a work (then a preprint) of the group of Estevez-Torres that was “a complementary, independent effort” to study similar instabilities.

That work has now already been published (ref 72 in the revised manuscript) and, from my perspective, has identical protocols, objectives and results using exactly the same active material. They even use a photoinduced transition between the two states using caged ATP. Granted, this is an irreversible transition (unlike the photogenetic material used here), but the idea, for the purpose of demonstrating the two states with the same material just by changing ATP concentration is the same. Interestingly, a similar note was added in ref 72 in regards of “a complementary, independent effort by the group of Guillaume Duclos who studied instabilities in kinesin microtubule active fluids”, citing this preprint.

After these observations, I make the two following considerations:

On the one hand, that note that the authors added after the acknowledgments of their original manuscript is obsolete and cannot be maintained in the same form in the revised manuscript, since ref 72 (or the preprint version) is now not a “last-minute surprise” anymore, as it is 6 months old at least, and it is a published paper.

On the other hand, that published paper will be indistinguishable, for most readers, from this submission.

In summary, I was ready to recommend acceptance of the manuscript in the current form, but, in view of my considerations regarding ref 72, I leave the final decision to the editors, as this is an editorial, rather than scientific, matter.

Reviewer #3 (Remarks to the Author):

In the revised version of the manuscript by Najma et al. and in the authors' response to the reviewers' comments, all my main concerns were addressed and resolved. The improved version of the text and figures clarifies the experimental observations and the approach they took for their experimental and theoretical study. The manuscript is clear, well written and represents an experimental advance in the field. I propose it for publication after a minor revision:

- In the meantime, between the first revision and this one, a similar paper was published (Sarfati et al). The authors cite it in the acknowledgements ([72]). Given the overlap in time of the submission of the two works, it is clear to me that this should not affect the novelty and originality of the manuscript. However, because of the similarities, I suggest mentioning and citing it in the introduction.

Another small suggestion:

- The labels of Fig. 5 are still wrong. In the text I read (lines 419-420):

Then, at low intensity, the network buckled out-of-plane (Fig. 5c). Increasing the light intensity induced a transition to an in-plane instability (Fig. 5d),

However, Fig. 5c corresponds to in-plane instability and Fig. 5d to the Blurriness graph.

In addition, two (b) points are indicated in the caption, one in line 464 and one in line 466.

We thank the reviewers for their comments. All revisions are marked in red in the revised manuscript. In particular, we removed any claims that observations of the bend instability suggest the active network is fluidized and emphasized that the active network is an active solid. We also addressed reviewers' comments about the theoretical model. Finally, we are now discussing the paper by Safarti et al in the introduction.

Response to reviewer 1

1. The authors say in this revised manuscript that ``Of note, both in and out-of-plane instabilities are possible in the original active nematic theory (36, 57) and have been reported in hydrodynamic simulations of purely extensile active nematic fluids (51). However, the out-of-plane component of the generic bend instability grows slower than the in-plane component when the active nematic fluid is confined in a thin channel (32). We are therefore confident that the bucking instability is a signature of the elasticity of the network. "

I am not sure how 32 establishes that a buckling instability requires network elasticity. That is, I don't see how 32 establishes that a fluid 2d layer (the theory that the authors construct assumes a 2d layer; that is, it averages over the thickness of the gel) necessarily bends in-plane before buckling. Note that the character of the in-plane and out-of-plane instabilities are somewhat different (and, consequently, the boundary conditions on the director field are probably different). The in-plane instability is analogous to the one discussed in 32: the instability of an active fluid filling a confined channel beyond a critical activity. The out-of-plane instability is distinct from this. It is an instability (at least in their calculation) in which the gel layer buckles as a whole without any extra distortion within the layer. There is no reason for that to grow slower than the in-plane bend instability. To see this, consider the following toy model: a 2d fluid nematic membrane. The nematic order is along x . The y confinement is smaller than the active lengthscale (this direction is perpendicular to the membrane normal, which is along z); therefore, there is no in-plane bend instability at all. However, as essentially the authors' calculation here shows, the buckling instability is still inevitable in this case (inevitable, if we assume a tension-free membrane). The height fluctuations of the membrane are still slaved to director fluctuations, leading to a negative effective surface tension along x . The only stabilisation of height fluctuations is due to the bending modulus, which appears at higher order in gradients (along x). Therefore, in this geometry, one has a buckling instability but no bend instability in a fluid system. The confinement along the z direction doesn't change this conclusion. Therefore, the blanket statement that the authors make is incorrect, and the buckling instability is not inherently a signature of the elasticity of the network.

When we mentioned `original active nematic theory', we meant the hydrodynamic model in reference 32, which is a 3D active nematic in a 3D channel, and by a thin channel, we meant a channel whose z dimension is much smaller than the x or y dimensions. In that model (which does not include any elasticity), we had shown that the out-of-plane component of the nematic

director grows slower than the in-plane component. However, the 'out-of-plane instability' in reference 32 corresponds to the growth of the out-of-plane component of the nematic director without an associated buckling instability and is different from the buckling instability discussed in this paper. In the limit of a fluid active nematic (i.e., no crosslinking) as in reference 32, the director can fluctuate freely without height fluctuations. Thus, we maintain that the out-of-plane buckling presented in the current manuscript is a signature of elasticity. However, we agree with the reviewer (see response to point 2) that the in-plane bending is not necessarily a signature of fluidization.

2. In the response letter, the authors say "We removed any claim of the transition from bending to buckling correspondsto the fluid to solid transition in the revised manuscript (see updated section I.304-I.309)." Yet, in the main text, the authors say "We hypothesize that the ATP-controlled transition from out-of-plane buckling to in-plane bending is a signature of a solid-to-fluid transition. (I188-9)". Later, they write "We performed over N=1000 experiments to build four experimental phase diagrams showing how the transition from an active fluid to an active solid depends on ATP, motor clusters, and crosslinker concentrations, and the average length of the microtubules (Fig. 3). "This seems contradictory. Possibly, the authors mean that they have removed all mention of solid-fluid transition in the theoretical section (which they have; they also correctly point out that there is no solid-liquid transition). However, the claims in the theoretical and experimental sections are somewhat contradictory. The claim in the experimental section is that there is a true solid-fluid transition, while the theory suggests that no such transition is actually necessary (the authors claim that this is consistent with all their observations). All one needs is the activity to be large enough or the shear modulus to be small enough. If that is the claim, the statements about the solid-fluid transition hypothesis have to be culled from other parts of the paper. Instead, one has the more consistent (but perhaps less spectacular) hypothesis that the shear modulus of an active elastomer changes with ATP. Note that these are, in principle, two distinct hypotheses and what sort of parameters controls the behaviour of the macroscopic gel depend on the hypothesis.

We agree with the reviewer. We have now modified these two sentences in the experimental section (see new I.189 and I.348).

3. I don't know why the authors label the bend instability a fluid-like instability and the buckling a solid-like instability. The authors themselves show that the bend instability exists in a system with a finite shear modulus (by definition, a solid). It is also evident that the buckling instability could have affected a fluid layer (as discussed above). I find this nomenclature confusing.

We can consider the solid and the fluid as two extremes of our parameter space. In the limit of high crosslinking and low activity, we have a solid that buckles out-of-plane, whereas in the limit of zero crosslinking and high activity (in which our theory is ill-defined), we have a fluid that bends in-plane. However, the transition from buckling to bending itself cannot be called a solid to fluid transition. Therefore, we have removed statements that lead to this confusion and emphasized that our system is an active solid [I2 (title), I.28 (in the abstract), I.92-94, I.148, I.189, I.238 (in the figure 2), I.312-313, I.348, I. 361, I.365, I.422].

4. I agree that the elastomer will be semi-soft and also that it is possible that the experimental regime is such that the semi-softness is not probed. I further agree that the surface tension (incidentally, it must be present and generically be non-zero -- since it is allowed by symmetry, its value cannot be 0 except at special points in the parameter space. Since the system is not tuned to a critical point with 0 surface tension, there will be a surface tension) may be small. However, the argument presented by the authors for the small value of the surface tension should be in the main text, and they should clearly say why they ignore surface tension in the main text.

We have now included this point in the main text (I.287-291)

5. The authors mention a theory involving the apolar order parameter Q_{ij} and the strain tensor u_{ij} in a footnote in the supplement. However, strangely, the energy they use for the nematic elastomer has a term $\gamma(\nabla \cdot \mathbf{u})Q_{ij}u_{ij}$ but not a simpler term $\propto Q_{ij}u_{ij}$ in the free energy. This is not inconsequential. A look at PRE 66, 011702 shows that the director is indeed slaved to the strain field but $\delta n_{\perp} \neq u_{\parallel}$ as the authors have but $\delta n_{\perp} \propto u_{\parallel}$ where the proportionality constant depends on the value and the sign of the coefficient of the $Q_{ij}u_{ij}$ free-energy coupling. That is, $\delta n_y \neq \partial_x u_y$ but $\delta n_y \propto \tilde{\gamma} \partial_x u_y$ where $\tilde{\gamma}$ is the coefficient of the $Q_{ij}u_{ij}$ free energy term. While the final free-energy (only depending on the displacement field) that the authors write can still be obtained from the nematic elastomeric theory, the consequence of the non-trivial slaving of the director field to the displacement fluctuations are important for the active force density. In particular, the coefficient (and sign) of the planar active force density should depend on the combination $\zeta \tilde{\gamma}$. The authors essentially seem to tacitly set $\tilde{\gamma}=1$. This needs to be justified, especially since $\tilde{\gamma}$ can, in principle, have an arbitrary sign (this is related to my earlier question "seems to claim that extensile active nematic elastomers are always unstable to bend at sufficiently high activity and, conversely, contractile active nematic elastomers are always unstable to splay at sufficiently high activity. Is that really the case?").

In our system, the only possible relationship between the nematic director and the strain tensor is $n=F.n_0$, where F is the deformation tensor, because the nematic director and the material that is deforming are one and the same. This situation is different from conventional nematic elastomers, 'composed of liquid crystalline mesogenic units covalently bonded to crosslinked polymer networks' (Eur. Phys. J. E 40, 76 (2017)), in which case the relationship between the nematic director and the strain tensor is determined by the free energy.

Nevertheless, to relate our work to earlier literature on nematic elastomers, we now mention in the SI that the relationship between the nematic director and the strain tensor that we use ($n=F.n_0$) can be obtained from the neoclassical theory of liquid crystal elastomers, by minimizing the free energy with respect to the nematic director under the assumption of strong crosslinking and high nematic order at the time of crosslinking (see Eur. Phys. J. E 40, 76 (2017)). This limit is also called frozen nematic director/nematic glass in the nematic elastomer literature.

The active force that we write down can be obtained by taking the functional derivative (wrt to u) of the term in the free energy that the reviewer mentions ($Q_{ij}u_{ij}$). Therefore, $\bar{\gamma}=\zeta$ is the activity by definition. This is the reason why we did not include this term in the free energy separately.

6. The authors in the revised manuscript talk about elastic moduli “below the isostatic transition”. I am confused by this. I thought that the elastic moduli vanish if the crosslink density is below the isostatic transition (there are zero modes and hence, the solid cannot resist any imposed stress).

We have fixed this typo to “above the isostatic transition”.

7. While the authors make a serious and commendable attempt to obtain the values of the hydrodynamic parameters used in active gel theories from the properties of microscopic constituents, because of various uncertainties (even for the active coefficient), they essentially only succeed in obtaining order of magnitude estimates. Other articles have attempted to obtain such order-of-magnitude estimates before (with varied success). Therefore, to me, stressing this too much sounds a bit like overselling.

We agree that our estimates are not quantitative and clearly mention the limitations of our approach in the discussions (see I.376-381, I.441 for the reference to the magnitude of the active stress, and I.442 for the uncertainty coming from the uncertainty on the scaling for the elasticity). We also removed from the abstract the claim that our estimates were quantitative.

Reviewer #2 (Remarks to the Author):

In this resubmission, Nahma et al. have significantly improved the manuscript, and have addressed my comments and criticisms in a satisfactory manner. In particular, they have clarified the difference with earlier work on, apparently, the same system. They explain that earlier work had a disparity of material conditions, which precluded a unified view. Instead, they demonstrate that different phenomena reported earlier can be replicated in the same system by changing the ATP concentration. This allows to propose a theoretical model to loosely explain the phenomena and, more importantly, to estimate important material parameters. In particular, the idea that this work presents a sort of “rheometer” to probe active material properties is quite attractive.

As a minor detail, I think there is an error in panel 5e (which escaped me in the original manuscript). The X-axis has units of time (it is a time), not a “rate”.

We fixed the typo in the figure 5e.

Finally, a major issue.

The authors were honest in their initial submission to acknowledge a work (then a preprint) of the group of Estevez-Torres that was “a complementary, independent effort” to study similar instabilities.

That work has now already been published (ref 72 in the revised manuscript) and, from my perspective, has identical protocols, objectives and results using exactly the same active material. They even use a photoinduced transition between the two states using caged ATP. Granted, this is an irreversible transition (unlike the photogenetic material used here), but the idea, for the purpose of demonstrating the two states with the same material just by changing ATP concentration is the same. Interestingly, a similar note was added in ref 72 in regards of “a complementary, independent effort by the group of Guillaume Duclos who studied instabilities in kinesin microtubule active fluids”, citing this preprint.

After these observations, I make the two following considerations:

On the one hand, that note that the authors added after the acknowledgments of their original manuscript is obsolete and cannot be maintained in the same form in the revised manuscript, since ref 72 (or the preprint version) is now not a “last-minute surprise” anymore, as it is 6 months old at least, and it is a published paper.

On the other hand, that published paper will be indistinguishable, for most readers, from this submission.

In summary, I was ready to recommend acceptance of the manuscript in the current form, but, in view of my considerations regarding ref 72, I leave the final decision to the editors, as this is an editorial, rather than scientific, matter.

We modified the acknowledgment section and are now citing and discussing the work by Sarfati and co-workers in our introduction.

Reviewer #3 (Remarks to the Author):

In the revised version of the manuscript by Najma et al. and in the authors' response to the reviewers' comments, all my main concerns were addressed and resolved. The improved version of the text and figures clarifies the experimental observations and the approach they took for their experimental and theoretical study. The manuscript is clear, well written and represents an experimental advance in the field. I propose it for publication after a minor revision:

- In the meantime, between the first revision and this one, a similar paper was published (Sarfati et al). The authors cite it in the acknowledgements ([72]). Given the overlap in time of the submission of the two works, it is clear to me that this should not affect the novelty and

originality of the manuscript. However, because of the similarities, I suggest mentioning and citing it in the introduction.

We are now citing and discussing the paper by Sarfati et al. in the introduction

Another small suggestion:

- The labels of Fig. 5 are still wrong. In the text I read (lines 419-420):

Then, at low intensity, the network buckled out-of-plane (Fig. 5c). Increasing the light intensity induced a transition to an in-plane instability (Fig. 5d),

However, Fig. 5c corresponds to in-plane instability and Fig. 5d to the Blurriness graph.

In addition, two (b) points are indicated in the caption, one in line 464 and one in line 466.

We revised the caption and fixed these typos.

REVIEWERS' COMMENTS

Reviewer #1 (Remarks to the Author):

In their revised manuscript, the authors have addressed most of my concerns. Some minor disagreements that I have should not prevent the publication of this article. I think the article can be published essentially as is.

The only suggestion I have is that the authors may consider changing the solid in the title to crosslinked gel